# Solving groundwater depletion in India while achieving food security

Naresh Devineni ⬡ [1,2✉], Shama Perveen[3] & Upmanu Lall ⬡ [2,4]

Significant groundwater depletion in regions where grains are procured for public distribution is a primary sustainability challenge in India. We identify specific changes in the Indian Government's Procurement & Distribution System as a primary solution lever. Irrigation, using groundwater, facilitated by subsidized electricity, is seen as vital for meeting India's food security goals. Using over a century of daily climate data and recent spatially detailed economic, crop yield, and related parameters, we use an optimization model to show that by shifting the geographies where crops are procured from and grown, the government's procurement targets could be met on average even without irrigation, while increasing net farm income and arresting groundwater depletion. Allowing irrigation increases the average net farm income by 30%. The associated reduction in electricity subsidies in areas with significant groundwater depletion can help offset the needed spatial re-distribution of farm income, a key political obstacle to changes in the procurement system.

[1] Department of Civil Engineering, City University of New York (City College), New York, NY 10031, USA. [2] Columbia Water Center, Columbia University, New York, NY 10027, USA. [3] Ceres, 99 Chauncy St. 6th Floor, Boston, MA 02111, USA. [4] Department of Earth and Environmental Engineering, Columbia University, New York, NY 10027, USA. ✉email: ndevineni@ccny.cuny.edu

Critical food shortages in India in the 1960s, including a major famine in 1965–66, led to the implementation of the Green Revolution and a reformulation of the Indian Government's food procurement and distribution system[1]. The procurement system was established with minimum price support for selected crops and a distribution/storage system to ensure that lower-income households have access to primary nutrition. To reduce transaction costs, the crops are procured from a few provinces where the Green Revolution took root and crop yields are typically higher. This ultimately led to a significant change in regional cropping patterns different from traditional crops adapted to the local climate and soil conditions. Increased demand for irrigation in the procurement regions resulted and surface water-based irrigation through dams and canals was developed in many cases but it had a limited reach. To support many smallholder farmers, state governments offered subsidized/ free electricity for pumping groundwater for irrigation which consequently led to widespread groundwater depletion[2,3]. Farmers in procurement-favored regions receive a guaranteed revenue (based on a margin above the cost of cultivation) from the crops procured by the government as well as subsidies for electricity and other inputs. This creates a political obstacle to change in the cropping system. Today, experts realize the distortionary impacts of the food procurement and distribution system on the cropping patterns and food prices. The reforms needed to delink these have been difficult to enact[4].

The spatial distribution of the 12 major crop varieties procured by the Central government in Kharif (June to September)—the predominant rainfall season[5] is shown in Fig. 1. Rice covers 75% of the net cropped area in the arid regions of North-Western India (e.g., Punjab and Haryana) and the Indo Gangetic Plains. These regions have low rainfall with high variability (Fig. S1) and hotspots for within-year and long-term chronic water stress[6].

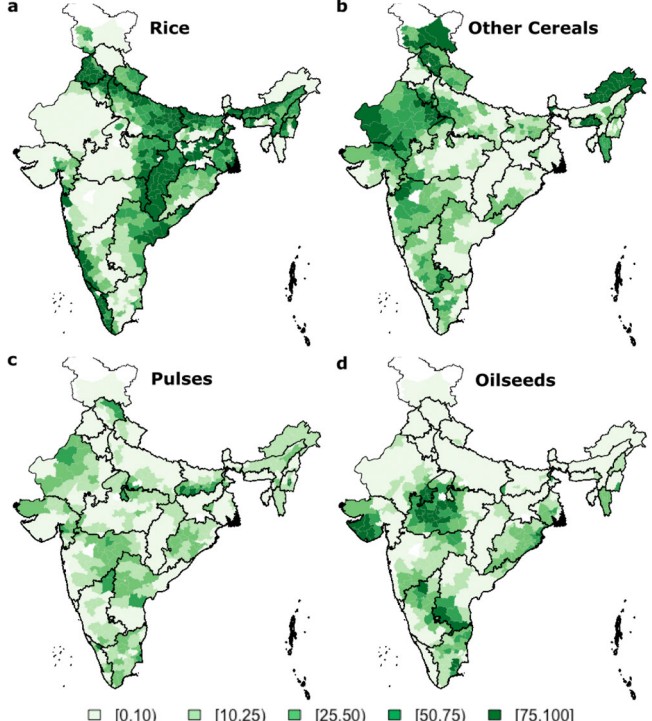

**Fig. 1 Current cropping pattern in India shown as percentage net-cropped area in Kharif season.** Twelve primary crops procured by the Ministry of Agriculture, Government of India, are grouped into rice (**a**), other cereals (bajra, maize, jowar, and ragi) (**b**), pulses (tur and other pulses) (**c**), and oilseeds (groundnuts, sesamum, soybean, nigerseed, and sunflower) (**d**).

Given subsidized electricity and limited access to canal water, most irrigation is from groundwater, accounting for as much as 40–60% of total electricity use in some states. Groundwater levels are dropping at rates between 1 and 3 m/year in these states[7], contributing to increasing costs for well deepening and pumps[8]. Socio-political and physical factors limit the degree to which irrigation efficiency improvements can address this situation[9].

There is recurrent interest in targeting new regions (e.g., Eastern India) for procurement[10,11]. A shift in cropping patterns and procurement strategies however needs careful analysis of climatic factors, economics, energy needs, and regional land type and crop productivity. A central question is whether a shift in grain procurement by the Government could economically achieve the national food security targets while addressing groundwater stress, the highly variable climate, and be economically feasible. Recently, Davis et al.[12,13] illustrated that India could improve water use and nutrition by shifting crops. This confirms our earlier results for purely rain-fed agriculture[14,15]. Bhogal and Vatta presented a meta-analysis of the studies that ascertain handicaps associated with crop diversification, especially for the state of Punjab[16].

Here, we provide a novel hydro-economic analysis suggesting possible entry points in designing the Indian government's food procurement system. Our model considers the maximization of net national farm income by allocating acreage per district for 12 major crops, accounting for weather/climate variations, regional crop productivity and cultivation cost, annual procurement targets and prices for each crop, and spatially specified limits on groundwater use. We consider two scenarios—(1) "Irrigation Zero" (i.e., rainfed agriculture), and (2) "Irrigation-Capped" (i.e., irrigated agriculture). Using the optimization model with these two scenarios and recent crop procurement prices (or the Minimum Support Price), we identify which crops should be grown and procured by the Government from each district to meet food security and nutritional needs while accounting climate variability over the last century. The integral consideration of district-level crop choice and productivity, climate variability, and economics provides an effective contribution to national policy goals that goes beyond prior related work.

## Results

**Research context**. From 2008 to 2015, the Columbia Water Center at Columbia University conducted several research studies and pilot projects in India to determine and address the factors underlying water stress due to climate variability and demands. Agriculture accounts for around 90% of water withdrawals in India. Groundwater is a primary source of drinking water, given the ephemeral, monsoonal rainfall systems that contribute to the renewable supply. Declining groundwater due to irrigated agriculture however also impacts costs and reliability of rural and municipal water supplies. Our detailed analyses of district-level water stress[6] have shown that the rice-wheat cropping system is a predominant contributor to the most severe groundwater depletion being observed across the country.

Three main solution directions, that cover a range of institutional and spatial scales, were considered to alleviate the current rapid groundwater decline: (i) Increasing farm-use water efficiency and water productivity[17], (ii) Conducting pilot projects to explore farmers' willingness to save water and electricity, including the state government's willingness for policy reform regarding water and electricity subsidies[18], and (iii) Restructuring the Government of India's procurement system to address the disproportionate impacts from the much needed but flawed Public Distribution System (PDS) in the country.

It was evident from the field experiments and farmer discussions that the critical dynamic determining farmer's crop choice was India's PDS. Even though it is national in scope, it targets only a few specific regions for crop procurement. The long history of guaranteed income to farmers in these regions and the development of supporting supply chains, markets, and agricultural extension programs undoubtedly make for a difficult change, as is evidenced by recent farmer unrest generated by the Indian government's recent proposed changes towards private markets. The guaranteed net income from the PDS system, the provision of free or subsidized electricity, and the Government's crop insurance schemes keep farmers from switching to alternate crops that could be more profitable. Consequently, for this analysis, we focused on targeting the Indian government's procurement system, which we believe is a primary driver triggering adverse crop selection, groundwater depletion, and high electricity usage (leading to agricultural subsidy-induced debts) in the country.

Geospatial data on district-level crop productivity, cost of cultivation, area under cropping and irrigated area, rural population demographics, minor irrigation infrastructure, and the Government's Minimum Support Price (base guaranteed price from the Government for the produce at the national level) and procurement levels were assembled from diverse sources (see Methods and the Tables in the Supplemental Material for details). These were then used in a crop allocation optimization model to identify areas where crops could be suitably grown in line with the available water, climate variability, and land-use types—to suggest possible entry points to help design PDS restructuring while ensuring food and nutritional security and maintaining financial viability for farmers across the country.

**Crop allocation optimization**. We developed a linear programming model with the objective of maximizing the expected value of the national agricultural net revenue at the published minimum support prices for each crop. The expectation or average is taken over the 1901–2009 period using the daily precipitation and temperature data to compute the potential crop yields at each location in the country for each growing season. Taking the district as a spatial modeling unit, the current cost of cultivation for each of the 12 PDS's MSP-supported crops was collected while accounting for the pumping costs. The model's decision variables are the fraction of the current cropped area allocated to each of the 12 PDS crops in each district. The constraints on the model include the following:

1. *Crop Demand Satisfaction at the national scale:* The average annual national production of each crop should exceed its current total production. This is to ensure that the consumption needs of the population are met.
2. *Nutritional satisfaction at the national scale:* The nutritional needs of the national population are met with the total production of all crops. This emphasizes a balanced diet rather than just meeting the food caloric needs, which had been the major driver for the predominance in cereal production (rice-wheat-maize) during the Green Revolution.
3. *Water Supply limit at the district scale:* Two scenarios were considered for irrigation. The total irrigation water and the irrigated area used in each district are limited to current levels ("*Irrigation Capped*") or to zero ("*Irrigation Zero*"). This ensures sustainable water use per district and avoids considering the cost of adding new irrigation infrastructure in places that are currently not equipped for irrigation. For the "Irrigation Zero" scenario, we were curious to see if it was possible to meet the PDS crop requirements with no

irrigation. The "Irrigation Capped" scenario acknowledges that farmers who have invested in irrigation may want to continue using it, as we seek the best crop allocation using irrigation across the country. Since the irrigated area in a district is often a fraction of the total cropped area, the "Irrigation Capped" scenario also considers non-irrigated agriculture at the same fraction of cropped area for each district.
4. *Land Use at the district scale:* Restriction of the cropped area per district to the current cropped area. This assures that we explore the possibility of crop substitution without increasing the total cropped area in any district, thus avoiding any deforestation or land-use changes.

Climate variability and its impact on the yield of Kharif season crops were considered using daily precipitation and temperature data covering 109 years (1901–2009) for both irrigation scenarios.

A detailed description of the model is provided in the Methods section. We identified the suitability of each crop for each district based on soils and climate along with any evidence that it is currently grown in that area. An extensive data set on district crop yields per unit area under "Irrigation Capped" and "Irrigation Zero" conditions was compiled from historical government survey data. Historical daily precipitation and temperature data from 1901 to 2009[19,20] were used to represent climate variability, its impact on crop water requirements, irrigation water requirements, and on the annual variation in the survey-reported crop yield at the district level, using Food and Agricultural Organization (FAO) methods[21]. The costs, nutritional contribution, and groundwater and energy use for the crops were estimated annually. Averaging over the 109-years climate scenario provided the average annual contributions to the net revenue function and to the constraints. We used only the Kharif (summer Monsoon) season for crops and irrigation in our analyses but used annual values for the procurement targets and nutrition.

**National farm income and food security**. For the "Irrigation Zero" agriculture scenario, we consider no irrigation across India. Crop yields for each district fluctuate based on the local daily rainfall pattern and water deficit with respect to the daily water requirement for each crop each year. The most remarkable conclusion is that just from the Kharif season, it is possible to choose a spatial cropping pattern that meets the annual PDS targets for crop production, even under the "Irrigation Zero" scenario, with a positive impact on net national net farm revenue.

A comparison of the current versus optimal aggregate values of the national revenue, production, and nutrition is provided in Fig. 2. The national agricultural net revenue (Eq. (7) (methods)) for the PDS crops generated from the proposed pattern for the "Irrigation Zero" scenario is INR 3.06 trillion, 5% greater than the revenue of INR 2.90 trillion generated from the current PDS cropping practice. Based on August 2020 Exchange rates, 1 USD = 75 INR. Net national revenue of INR 3.74 trillion is generated from the "Irrigation Capped" scenario, ~30% higher than the current revenue.

The net national agricultural revenue under the "Irrigation Zero" scenario increases through higher oilseeds production while meeting the current minimum production for rice, cereals, and pulses. Under the "Irrigation Capped" scenario, the net national agricultural revenue is increased by increasing the tonnage of pulses and oilseeds while meeting the other crops' minimum quantity. The nutritional value derived from the new national PDS production meets or exceeds the recommended intake and is greater than the current nutritional intake. The

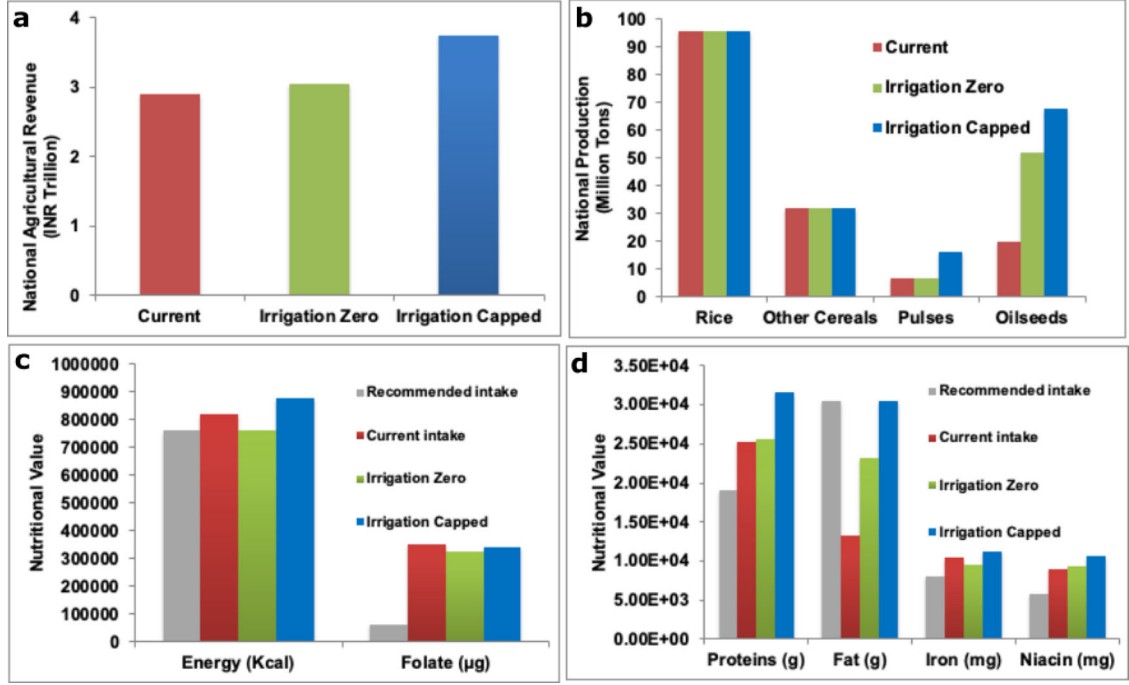

**Fig. 2 Comparison of current vs. optimal national revenue, crop production, and nutritional value. a** Current vs. optimal national agricultural revenue. **b** Current vs. optimal national production levels. **c** Current vs. optimal nutritional value for energy and folates. **d** Current vs. optimal nutritional value for proteins, fats, iron, and niacin. For nutritional value, the recommended intake is also shown.

caveat is that it does require the government to procure more pulses and oilseeds than it currently does.

**Optimal cropping patterns, water, and energy use**. The spatial distribution of the optimal cropping pattern under the two water supply scenarios is shown as a percent increase or decrease relative to the existing net-cropped area for each crop type in each district, in Fig. 3 (please refer to Fig. S2 for a map with Indian state names). Under both scenarios, Northern, Central Northeastern, and parts of Southern India emerge as the ideal locations for cultivating rice. The cultivated rice area in the northern arid states of Punjab, Haryana, and the Indo-Gangetic Plain, where over 50% of the PDS procurement occurs, is reduced by over 75%. It is interesting, but not surprising, to note that these current sourcing regions are reduced significantly under both scenarios, despite their relatively high productivity, reflecting the water supply imbalance that is driving groundwater depletion and high subsidized energy use in these regions.

Other cereals (jowar, bajra, maize, and ragi), which require less irrigation compared to rice, emerge as the crops best suited in the northern states of Punjab and Haryana, southern and eastern Andhra Pradesh (undivided as of 2001), and Chhattisgarh. This is the case even given their lower yields per unit area in these regions. Pulses increase in the districts in the Indo-Gangetic Plain. Similarly, for oilseeds, Rajasthan, Gujarat, Maharashtra, Orissa, and Tamil Nadu emerge as better cultivating locations. These "optimal" cropping patterns are broadly consistent with the cropping patterns that existed in the Northwest and Indo-Gangetic Plains before the Green Revolution and before the Government of India's PDS procurement strategy (private conversations with farmer networks). This confirms that the current PDS strategy inadvertently created an anomaly that contributes significantly to water resource stress. The cropping pattern that had traditionally existed was better optimized to the regional variations in climate.

We performed a sensitivity analysis on the net unit revenue (i.e., support price—the unit cost of cultivation) for the different crops and found that the optimal spatial allocation for rice, other cereals, and oilseeds is robust to these changes. The spatial crop allocation pattern from the optimization model does not change even if net unit revenue for each of these crops was reduced one at a time by 10–50% (see Fig. S3 in the supplemental material). However, the spatial allocation of pulses is sensitive to the net unit revenue, suggesting that the government should give careful consideration to the minimum support price offered for pulses to ensure proper targeting of procurement areas. Non-economic factors such as traditional crop choices in a region may be a factor in the allocation in addition to the support price. The current cost of cultivation for some states to which rice is moved (for example, over INR 700 per 100 Kg in Madhya Pradesh, West Bengal, and the Northeastern States), is higher than the cost of cultivation for rice in an arid state (e.g., INR 416 per 100 kg in Punjab) from which rice is moved by the model. This reflects a difference in rice yield (e.g., 4.5 tons/hectare in Punjab to 1.5 tons/hectare in some of the other states) due to agricultural practices. The crop shift despite these yield and cost differences reinforces the importance of the water constraint in the model. Technological innovations (e.g., appropriate cultivars, fertilizer, and harvesting practices) could reduce the cost of cultivation or equivalently increase the yield per unit land) making the shift even more attractive. Labor markets in some of these states may already be more attractive than in states like Punjab that import most of their farm labor.

At the national level, the "Irrigation Capped" agricultural scenario used slightly less irrigation water (130 Billion $m^3$) compared to the one based on the current cropping pattern (146 Billion $m^3$). The "Irrigation Zero" scenario uses no irrigation water, saving 146 Billion $m^3$. Correspondingly, the total aggregate national-level energy usage under the "Irrigation Zero" ("Irrigation Capped") agricultural scenario is 0 (25797) GWh, compared to the current usage of 26252 GWh at the national level.

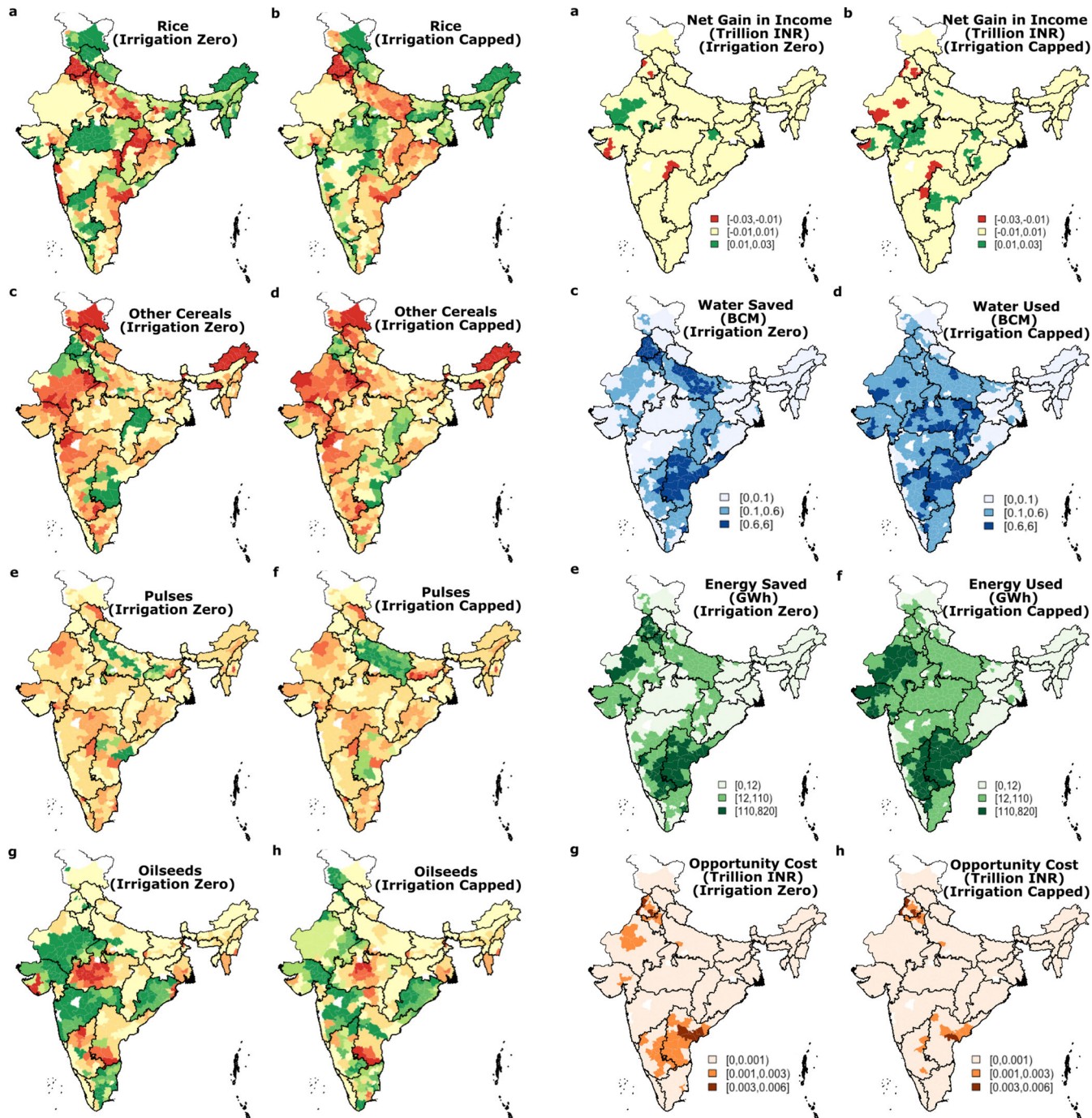

**Fig. 3 Changes to the cropping pattern in India are shown as percentage increase or decrease relative to the current cropped area in Kharif season under the two proposed scenarios. a**, **b** rice under the "Irrigation Zero" and "Irrigation Capped" scenarios. **c**, **d** other cereals under the "Irrigation Zero" and "Irrigation Capped" scenarios. **e**, **f** pulses under the "Irrigation Zero" and "Irrigation Capped" scenarios. **g**, **h** oilseeds under the "Irrigation Zero" and "Irrigation Capped" scenarios.

**Fig. 4 Spatial distribution of net revenue, water and energy use, and energy opportunity cost. a**, **b** Net gain in farm income under "Irrigation Zero" and "Irrigation Capped" scenarios. **c**, **d** Water saved/used under the "Irrigation Zero" and "Irrigation Capped" scenarios. **e**, **f** Energy saved/used under the "Irrigation Zero" and "Irrigation Capped" scenarios. **g**, **h** Opportunity cost under the "Irrigation Zero" and "Irrigation Capped" scenarios.

**Redistribution of income, water, and energy use from changing PDS**. A key concern with any proposed shift in the PDS is how the distribution of income may change since many of the farmers originally changed their crops because of the incentives offered by the government—a minimum guaranteed price to procure the crops that were above the average cost of production in the district. Electricity subsidies for groundwater pumping are typically

provided by the state governments and can account for a significant budgetary outlay (e.g., Punjab spent INR 65 billion in 2019–2020[22]). Water management is also a state responsibility in India. Consequently, the implications of the PDS change on farmer income, water, and energy are important to consider at the state level.

The distribution of net gain in farm income under both irrigation scenarios is shown in Fig. 4. For each district, we

compute the net change in revenue for the scenarios relative to the current cropping pattern. In Fig. 4, we also present the groundwater and energy saved (under the "Irrigation Zero" scenario) and used (under the "Irrigation Capped" scenario) in each district. The state-level aggregates of the same are shown in the supplemental information Fig. S4. We observe the spatial distribution of the net gain in income (first panel in Fig. 4 shows net gain in income in trillion INR in three categories—negative (−0.03 to −0.01), neutral (−0.01 to 0.01), and positive (0.01 to 0.03)—that for most of the country, under both scenarios, the income gains are in the neutral category. The neutral category is approximately ±6% of the change in net income at the national level (the current national agricultural income is 2.9 trillion INR and the income from rainfed scenario is 3.06 trillion INR, a net gain of 0.16 trillion INR).

The water and energy saved maps reveal that under the "Irrigation Zero" case, there is a significant saving in water and energy in the currently stressed regions of Punjab, Rajasthan, Uttar Pradesh, Andhra Pradesh/Telangana, and Karnataka. These are also the major regions that have a significant opportunity cost when we compared the savings in energy to current residential electricity price of 7 INR/kWh.

The state-level aggregates (presented in Fig. S4) are also worth discussing. For example, take the case of Punjab, which experiences significant groundwater depletion linked to rice cultivation for the PDS in the Kharif season. Under both scenarios, the nationally optimal crop allocation would translate into a net loss of farm income in Punjab of 86 and 79 billion INR, respectively. However, 38 (34) billion cubic meters of irrigation water and 4366 (3923) GWh of energy are saved annually relative to the current cropping pattern under the new "Irrigation Zero" ("Irrigation Capped") scenarios. The value of the energy savings alone is 31 (27) Billion INR at the current residential electricity price of 7 INR/kWh. Thus, while one can categorize most districts in Punjab in the neutral category of income change, even accounting for the value of the energy savings, one can see that Punjab would face a financial shortfall. The value of sustaining the groundwater resource in the state and reversing the 1–2 m/year rate of depletion would need to offset this financial challenge.

The growing recognition of the problem is motivating the state government to actively consider crop diversification, creation of a water authority, and experiments to pay farmers to conserve electricity. Past experiments to introduce private sector contract farming in the State have met with limited success as market prices for tomatoes and other perishables procured by the companies fluctuated, prompting reneging contracts by farmers and companies[23–25]. Consequently, farmers prefer the assured PDS scheme. Therefore, one would need to find a way to introduce a crop pricing and procurement system that would be politically and economically favorable to farmers to stimulate a transition towards growing alternate crops. Developing a robust supply chain for higher-value fruits, vegetables, and dairy with price guarantees and preservation would make sense for a state like Punjab to overcome the financial challenge it faces currently. It would contribute to India's nutrition and agricultural exports.

There are also winners. Under the "Irrigation Zero" ("Irrigation Capped") scenario, 12 (19) out of the 32 states increase net revenue. Significant net gains in net farm income under the "Irrigation Capped" scenario are seen in Andhra Pradesh (99 billion INR), Chhattisgarh (90 billion INR), Gujarat (141 billion INR), Madhya Pradesh (239 billion INR), Orrisa (136 billion INR), and Uttar Pradesh (281 billion INR). Only four states have a sizable loss compared to their current revenue. These include Maharashtra (111 billion INR), Punjab (79 billion INR), Karnataka (75 billion INR), and Haryana (21 billion INR). Groundwater depletion is a significant issue in Andhra Pradesh,

Gujarat, Punjab, Karnataka, and Haryana. Targeting these states for a new agricultural supply chain development that is water sensitive and yields higher net income beyond the choice of crops considered here would then be indicated as a goal for a state or nationally sponsored scheme like PDS.

Fruits, nuts, and vegetables are expensive in India and would potentially use less water if efficiently irrigated. However, they experience significant price fluctuations, and no robust supply chains that promise guaranteed revenues to farmers for these perishable commodities have emerged at a large scale. These suites of crops may provide an opportunity for the four states that do not benefit financially from PDS restructuring but have evolved to substantially higher agricultural productivity relative to the states that look favorable for PDS development for the existing crops procured.

## Discussion

India's PDS can be seen as a version of contract farming, in which a farm producer and a buyer agree on terms for the quality, quantity, and price for what is to be grown. The Indian Government has demonstrated that this is a powerful tool for shaping crop choices and integrating technology, financing, appropriate inputs, and insurance instruments to support rural economic development while meeting the food security and nutritional needs of 1.3 billion people. It is an extraordinary success story for a country plagued by famine when its population was 300 million. Over time, the intensification and concentration of agriculture to support the PDS goals have led to poignant environmental and water resource sustainability concerns.

Areas with a high concentration of PDS procurement have no doubt benefited through economic development but also face problems, with concerns about the financial viability of state governments who seem to be trapped in providing increasing subsidies for electricity and groundwater pumping, and of the long-term viability of agriculture as natural resources get depleted. In contrast to the general analyses of water, energy, and food in India, in this paper, we focused deliberately on the role of the PDS and the opportunity for its reform to address this situation. It is remarkable that a single lever at the command of a government can be tweaked to achieve a significant change in a nation's water, food, and energy outcomes. Its effectiveness with the objective of achieving self-sufficiency in grains has been well demonstrated. Its potential for addressing water and energy sustainability while addressing the original objectives and accounting for the economics of transition is seen to be promising from our analysis. The politics and economics of rural development and agriculture are complex. Consequently, an analysis of the type presented here is important in shaping some attributes of that discussion. We have anticipated some of the factors that could emerge and have discussed them. Feedback on our results will stimulate modifications of the analysis that may provide more insights. We expect that a multi-objective formulation that considers equity across states relative to water and production risks in addition to net farm revenue could be useful. Additional instruments for financial risk management and solar-powered shallow groundwater irrigation development could be considered in this case. The analysis of shadow prices of the constraints (presented in Fig. S5 in the supplemental material) could serve as the basis for such economic discussions.

At the level of impact, our discussions with farmers in Punjab and Gujarat indicate that they would welcome returning to traditional (or other) crops grown in those areas, with government procurement at a cost-plus margin (the current PDS model). There is an expectation that the government would also support these measures through research, agricultural extension,

technology, and financial measures to increase the productivity and reliability of production of the alternate crops. Thus, a refocusing of the PDS could indeed allow India's Government to facilitate a transition to a sustainable water-energy-food future. In our analysis, we accounted for the historical climate variations. Climate change projections for the 21st century continue to be somewhat uncertain. Consequently, an adaptive strategy that considers 5-year plans and associated climate change scenarios would provide a robust strategy for moving toward a sustainable future. Our future research endeavors will continue to address these.

## Methods

### Data

*Climate and groundwater data.* Rainfall data at the daily time scale from 1901 to 2009 (109 years), and at a spatial resolution of $1^0$ by $1^0$ are available from the Indian Meteorological Department (IMD)[19]. Temperature data at the daily time scale from 1969 to 2005, and at the same spatial resolution of $1^0$ by $1^0$ are also available from IMD[20]. We used simulated data from daily climatology (mean and standard deviation of daily temperature) as a proxy for the 1901–1968 and 2006–2009 spans to augment the temperature data and match the time period of the rainfall data. From the daily minimum, mean and maximum temperature, and the latitude and extra-terrestrial solar radiation, we computed the daily reference crop evapotranspiration ($ET_0$) for each of the $1^0$ by $1^0$ grid using the Hargreaves method[26]. The Hargreaves method for estimating $ET_0$ is typically employed in regions where data availability is limited to air temperature[21]. The $1^0$ by $1^0$ climate grids are spatially interpolated to 586 districts in India based on a 2001 district boundary layer to create a district level, 109-year daily time series data of rainfall and $ET_0$. We then use this district-level rainfall and $ET_0$ data to estimate crop-specific water deficit.

India's Central Ground Water Board (CGWB) estimates groundwater extraction using a number of wells and a uniform assumption on the extraction of each type of the well[27,28]. The average depth to groundwater level for each district is computed from this data. State-wide percentage coverage of irrigated area for major crops is available from the Directorate of Economics and Statistics, Ministry of Agriculture, India—DACNET[5].

The district-level average annual rainfall, its inter-annual coefficient of variation, the average depth to groundwater level estimated based on the CGWB data, and the state-wide percentage total irrigation area coverage under all the crops are shown in Fig. S1 of the supplemental material.

*Agricultural data.* The Directorate of Economics and Statistics, Ministry of Agriculture, India[5] hosts the Indian Harvest Database. We selected twelve MSP-supported primary Kharif season crops and obtained their respective cultivated areas and yields at the district level. The Kharif season spans from June to September, and is the predominant rainfall season that accounts for 90% of the annual rainfall. We grouped the crops into cereals (rice, bajra, maize, jowar, and ragi), pulses (tur and other pulses), and oilseeds (groundnuts, sesamum, soybean, nigerseed, and sunflower). The cereals and pulses selected here together comprise about 98% of the total food grains produced in the Kharif season[5]. The four chosen oilseeds account for about 93% of the total oilseeds produced in the Kharif season.

For each crop variety, we determine the potential yield under experimental conditions. For the potential yield for cereals, we used national average yields based on-farm research demonstrations over 13 years[29]. For pulses, we used the potential yields reported by the Expert Committee Report on Pulses (TMOP)/MOA[30]. Since we were unable to obtain such estimates for oilseeds, we used the maximum actual yields across all districts over the past 15 years as the potential yield for oilseeds if full irrigation was applied.

We also accessed the current seasonal agricultural production, the minimum support prices, and the cost of production for each of these crops as of the 2018 Kharif season. The cost of production covers all the tangible expenses incurred by the owner, i.e., the interest on the value of owned lands and fixed capital assets; the rental value of owned land, and credited value of fixed capital assets in addition to the direct costs (seeds, fertilizers, irrigation, labor, etc.)[5]. The cost of production data is available at the state level. In this study, we used the average of the previous three cropping years (2013–2014, 2014–2015, and 2015–2016) as the estimate for the cost of production. For the states where this data is not available, we assume a national average per crop. Further, we assume the same cost for all the districts in a state. The Ministry of Agriculture of the Government of India announces the minimum support price (MSP) at the beginning of each season based on the Commission for Agricultural Costs and Prices recommendations. All these details are provided in Tables S2 and S3 of the supplemental material.

The 2001 estimates of the population for each district are obtained from the Census of India[31]. The Indian Census data on population is available every ten years beginning 1872.

The nutrient composition of each crop variety is obtained from the United States Department of Agriculture (USDA), National Nutrient database[32]. The USDA Nutrient Database is a major source of food composition data in the United States and has information for 7636 food items. The recommended daily intake of nutrients in the diet for various groups of people, particularly in developing countries are obtained from the Food and Agriculture Organization of the United Nations (FAO) database. It provides safe levels of intake for a variety of nutrients for different gender and age groups. Safe levels of consumption are the levels that maintain health and nutrient stores in healthy individuals within a group. Further, these recommended intakes provide sufficient amounts of nutrients for prevention of deficiency disease, for growth and healthy maintenance of the body, and optimum levels of activity[33]. These details are provided in Table S4 of the supplemental information.

### Models

*Estimating crop water deficit and yields.* For each crop, we first calculate the Kharif season crop water deficits using the daily 109-year time series of rainfall and $ET_0$. The deficit, estimated as the difference between daily potential crop water requirement and renewable water supply is accumulated over the entire season. The maximum accumulated deficit over the season is considered as seasonal crop water deficit that may lead to a reduction in crop yield if irrigation is not provided.

A fraction of daily rainfall is assumed to be available as water supply for each day.

$$S_{j,t,d} = \alpha * P_{j,t,d} \qquad (1)$$

$P_{j,t,d}$ is the rainfall over a district $j$, for an year $t$, and a day $d$. $\alpha$ is the parameter that determines the fraction that can be utilized for crops. For this analysis, we set $\alpha$ at 0.7[6].

For each crop, we estimate the daily water use based on the expected growth stage and evapotranspiration.

$$D_{i,j,t,d} = [k_c]_{i,d} * [ET_0]_{j,t,d} \qquad (2)$$

$[k_c]_i$ is the crop coefficient for crop $i$. It is the ratio of the actual evapotranspiration ($ET_a$) under non-stressed conditions to the reference crop evapotranspiration ($ET_0$). It represents crop specific water use at various growth stages of the crop, and is typically derived empirically based on local climatic conditions[34]. The total crop water requirement over the entire season of $n_s$ days (approximately 120 days) is:

$$CWR_{i,j,t} = \sum_{d=1}^{n_s} D_{i,j,t,d} \qquad (3)$$

The accumulated deficit over a season is given as:

$$deficit_{i,j,t,d} = \max\left(0, deficit_{i,j,t,d-1} + D_{i,j,t,d} - S_{j,t,d}\right), where\ deficit_{i,j,t,d=0} = 0 \qquad (4)$$

and the seasonal crop water deficit is:

$$SD_{i,j,t} = \max_d\left(deficit_{i,j,t,d}\right) \qquad (5)$$

The seasonal crop water deficit ($SD_{i,j,t}$) focuses on rainfall distribution within the season relative to the crop water demand. It, therefore, accounts for the timing of planting, different stages of crop growth, and the timing and distribution of rainfall in the season, and hence, can discriminate between 2 years that have the same total rainfall but differ in their daily distribution. For instance, one year may have rainfall distributed uniformly over the season through modest rainfall events, while the other may have a few intense rain events separated by extended dry periods. The latter has an immediate and adverse effect on the crop growth and hence the yield.

Using $SD_{i,j,t}$, the seasonal crop water deficit, we estimate the crop yield.

$$Y_{i,j,t} = \left(1 - \frac{(1 - \eta_{i,j}) * SD_{i,j,t}}{CWR_{i,j,t}}\right) * PY_i \qquad (6)$$

$SD_{i,j,t}$ and $CWR_{i,j,t}$ are the seasonal crop water deficit and the total crop water requirement estimated for each crop $i$, in a district $j$, for a year $t$. $\eta_{i,j}$ is the irrigation potential for crop $i$, and district $j$. $PY_i$ is the potential yield for crop $i$, defined as the yield attained when cultivated under favorable conditions with full irrigation and nutrient supply. $PY$ is the maximum achievable yield for the crop under non-stress conditions. For $\eta_{i,j}$, we used the state-wide percentage coverage of irrigated area for these crops that is available from DACNET[5]. Details for all the states are provided in Table S5 of the supplemental information. We use the maximum fraction per district as the irrigation potential for all the crops in that district. As an example, for the districts in Punjab, the percent area irrigated under rice and maize are 97% and 64%, respectively. In the optimization model, we assume that all the crops in Punjab can be irrigated up to 97%. This fraction of the seasonal crop water deficit can be supplied through irrigation, and hence, if the irrigation potential is close to 1, the estimated yield $Y_{i,j,t}$ approaches potential yield $PY_i$. The expected value of the estimated yield is used in the crop allocation model.

*Crop allocation model.* Our crop allocation model is developed using linear programming. With an objective to maximize the aggregate national agricultural revenue, the model determines feasible regions and crop choices across India for the Kharif season while trying to satisfy a set of linear constraints.

We define aggregate national agricultural revenue as the difference between the expected value of the total income from all the crops cultivated in the season in all the districts and the cost of cultivation of these crops, including the cost of irrigation. We impose the following constraints on the model.

1. A district-level upper bound on the total cropped area.
2. A district-level upper bound on the total irrigation water.
3. A national food security constraint in terms of target production of the crops.
4. Target nutritional requirements recommended for the entire population.

In addition to the district-level irrigation constraint, Eq. (6) also serves as an implicit water sustainability constraint in the model. As explained in the previous section, we estimate crop yield as the reduction from potential yield due to crop water deficit that cannot be supplied through irrigation. Hence, in districts where irrigation potential is close to zero, yield loss resulting from crop water deficit is high for crops that require more water through the season (e.g., rice) compared to crops that require less water through the season (e.g., pulses). Consequently, the annual revenue generated from a crop with high water requirements in a district is lower than the revenue generated from a crop with low water requirements. Further, yield loss that results from crop water deficit is high for districts in arid regions that cannot provide irrigation than districts in a humid region. Hence, the model would identify suitable crops for districts per their climatic patterns.

The model is formally presented below.

Objective function: The goal is to maximize the expected net national agricultural revenue

$$O = \mathop{E}_{t}\left[\sum_{j=1}^{n_d}\left(\begin{array}{c}\left(\sum_{i=1}^{n_c}\delta_{i,j}*\left(MSP_{i,j}-CP_{i,j}\right)*Y_{i,j,t}*a_{i,j}\right)\\ -CI_j*\left(\sum_{i=1}^{n_c}\left(\frac{1}{\beta_i}*\eta_{i,j}*SD_{i,j,t}*\delta_{i,j}*a_{i,j}\right)*\psi_1*g*h_j*\frac{1}{\mu_p}\right)*\psi_2\end{array}\right)\right] \tag{7}$$

$\delta_{i,j}$ is the indicator function that determines the suitability of crop $i$ in district $j$. While this is typically determined using soil characteristics and temperature profile, we estimate this based on the historical crop cultivation data in this study. If crop $i$ was cultivated in district $j$ for at least five times in the past, we assume that the district is suitable for this crop—$\delta_{i,j}=1$. $MSP_{i,j}-CP_{i,j}$ is the net profit (INR/kg) resulting from crop $i$ in a district $j$. $MSP_{i,j}$ and $CP_{i,j}$ are the minimum support price and the cost of cultivation, respectively. These returns can be based either on the government announced minimum support prices, which are constant across the whole country, or the market prices, that can vary by the district. The cost of production typically varies by crop across the country. $Y_{i,j,t}$ represents the yield (Kg/Ha) estimated from crop water deficit for crop $i$ in district $j$ for a year $t$ (see Eq. (6)). $a_{i,j}$ is the decision variable i.e., the area (Ha) allocated for each crop $i$, in district $j$. $CI_j$ is the electricity cost charged for irrigation. We assumed a nominal national flat charge of INR 3/kWh. The average agricultural power tariff in 2011 was around INR 1.5/kWh[18,35]. The term $(\frac{1}{\beta_i}*\eta_{i,j}*SD_{i,j,t}*\delta_{i,j}*a_{i,j})$ is the total irrigation water pumped for crop $i$ in district $j$. It includes an irrigation efficiency factor $\beta_i$ to adjust for additional losses due to application efficiency. For rice, we assumed a 30% efficiency (due to its flood irrigation practice). For the other 11 crops, we assumed a 75% irrigation efficiency[36]. $\psi_1$ is the conversion factor from volume to mass. Since $SD_{i,j,t}$ is in units of millimeters, and $a_{i,j}$ is in units of hectares, $\psi_1 = \frac{1}{1000}(m)*10,000(m^2)*1000(\frac{kg}{m^3})$. $g$ is the acceleration due to gravity on earth, 9.81 m/s². $h_j$ is the average depth (in meters) to groundwater level in district $j$ from where water is extracted for irrigation. District-level data for average depth to ground water levels are available from the Central Ground Water Board (CGWB). $\mu_p$ is the coefficient to account for pump efficiency. We assumed that pump efficiency in all the districts is 30% based on the efficiencies reported in various Indian states[37,38]. Finally, $\psi_2$ is the conversion factor from Joules to kWh—(1/3600,000).

The operator $\mathop{E}_{t}[.]$ denotes the expectation of the objective function, and $n_c$ and $n_d$ are the number of crops for the season (12) and the number of districts (586) in the country, respectively.

Constraints: We group the constraints into three categories: (a) area and location constraints, (b) irrigation constraints, and (c) food security and nutritional constraints.

The area and location constraints prescribe the maximum area allocated for agriculture in a given district and the suitability of the type of crop in that district.

$$0 \leq \sum_{i=1}^{n_c}\delta_{i,j}*a_{i,j} \leq TCA_j \; \forall j \tag{8}$$

$TCA_j$ is the total Kharif season cropped area for the selected crops in each district $j$. The area and location constraint ensure that the allocated crop acreage is within maximum possible cropped area in a given district.

The irrigation constraint ensures a sustainable limit—it is restricted to be no more than 15% of the average annual rainfall. We assume that 15% is the percentage of average annual precipitation that recharges groundwater, a

reasonable assumption for subhumid to humid regions[39,40]. This quantity is available as renewable groundwater.

$$0 \leq \mathop{E}_{t}\left[\sum_{i=1}^{n_c}\frac{1}{\beta_i}*\delta_{i,j}*\eta_{i,j}*SD_{i,j,t}*a_{i,j}\right] \leq \lambda*\bar{P}_j*A_j \; \forall j \tag{9}$$

$A_j$ is the net cropped area, and $\bar{P}_j$ is the average annual rainfall for district $j$. We set $\lambda = 0.15$.

The food security constraint ensures that the aggregate produce from different crops is at least as much as the current aggregate produce.

$$\mathop{E}_{t}\left[\sum_{j=1}^{n_d}\delta_{i,j}*Y_{i,j,t}*a_{i,j}\right] \geq Q_i \; \forall i \tag{10}$$

$Q_i$ is the current national aggregate production of crop $i$. The number of food security constraints will be equal to the total number of crops chosen. This constraint ensures that the net agricultural produce resulted from the new allocation is at least equal to the current net production of each of these crops.

Lastly, we introduce nutrition targets since self-sufficiency in terms of aggregate food grains produced does not ensure nutritional goals. Our nutrition constraints ensure that the total nutritional requirement for a selected spectrum of nutritional goals is at least as much as the recommended nutritional goals for the population.

$$\mathop{E}_{t}\left[\sum_{i=1}^{n_c}\sum_{j=1}^{n_d}\delta_{i,j}*c_{ni}*Y_{i,j,t}*a_{i,j}\right] \geq N_n \; \forall n \tag{11}$$

$N_n$ are the nutritional needs of the country's population corresponding to a suite of nutritional goals ranging from calories, proteins, fats, etc. $c_{ni}$ is the amount of nutritional content for nutrient $n$ (calories, proteins, etc.) in crop $i$.

This model has $(2n_d + n_c + n)$ number of constraints and can be solved using any of the traditional linear programming algorithms such as the simplex algorithm[41]. We used the simplex algorithm available through the *lpSolve* solver package in R[42].

*Scenarios.* We considered two scenarios, "Irrigation Capped" and "Irrigation Zero". The "Irrigation Capped" scenario considers irrigation and has the following constraints: (a) area and location constraints, (b) irrigation constraints, and (c) food security and nutritional constraints. Here we assumed $\eta_{i,j} = \max_i(\eta_{i,j})$, i.e., for each district, the irrigation potential for any crop is the maximum irrigation potential in that district. For example, for districts in Punjab, the percent area irrigated under rice is 97%, the largest for any crop in Punjab. We assume that any crop in Punjab can be irrigated to this level. The "Irrigation Zero" scenario considers no irrigation. Here, the model has only area, location, food security, and nutritional constraints. We assumed no irrigation potential for the country, i.e., $\eta_{i,j} = 0$ for all the crops and districts. For the "Irrigation Capped" scenario, 1190 constraints (586 district area constraints; 586 district irrigation constraints; 12 production constraints; six nutritional constraints—energy, proteins, fat, iron, niacin, and folate) result. For the "Irrigation Zero" scenario, 604 constraints (586 district area constraints; 12 production constraints; six nutritional constraints—energy, proteins, fat, iron, niacin, and folate) need to be satisfied.

## Data availability

The rainfall and temperature data used in this study are available from the Indian Meteorological Department: https://cdsp.imdpune.gov.in/home_gridded_data.php District-level data for average depth to groundwater levels is available from the Central Ground Water Board (CGWB): http://cgwb.gov.in/GW-data-access.html The crop-relevant data are available from the Directorate of Economics and Statistics (DACNET): https://eands.dacnet.nic.in. They are also provided in the tables (Tables S2–S5) in supplementary information file. The 2001 estimates of the population for each district are available from the Census of India: https://censusindia.gov.in/census.website/data/census-tables All data used in the study are also available from the corresponding author upon reasonable request.

## Code availability

The code developed in the current study is available from the corresponding author on reasonable request.

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

## Acknowledgements

Support from the PepsiCo Foundation (Project Title: *Improving rural water and livelihood outcomes in India, China, Africa, and Brazil*) is gratefully acknowledged. The work presented in this paper resulted from a series of exchanges with water and agricultural experts, including decision-makers in India. We particularly thank Drs. Rajinder Sidhu, Kamal Vatta, and Baljinder Kaur from the Punjab Agricultural University, Dr. S.S. Johl, ex-Vice Chancellor, Punjabi University, Dr. Ram Fishman from Tel Aviv University, Dr. Vijay Modi from Columbia University, Dr. Kapil Narula from CII-Triveni Water Institute, Dr. Alok Sikka from IWMI, Raman Ahuja, ex-Head of Strategy, Field Fresh, Sanjeev Chaddha, ex-CEO, and Vivek Bharti, Executive Director Corporate Affairs, Pepsico India, Sanjeev Asthana, ex-CEO Reliance Retail and Dr. Montek Singh Ahluwalia, ex-Deputy Chairman, Planning Commission of India. The feedback from these experts was invaluable. The opinion, findings, conclusions, and recommendations expressed in the article are those of the authors and do not reflect the views of the organization they are affiliated with.

## Author contributions

All authors conceived the idea. N.D. and U.L. formulated the optimization model. N.D. and S.P collected the data. N.D. performed the analysis. N.D. and U.L. wrote the text. All authors reviewed and edited the manuscript.

## Competing interests

The authors declare no competing interests.
