## [Peer Review File · Nature Communications]

Solving Groundwater Depletion in India while achieving Food SecurityReviewers' Comments:

Reviewer #1:

Remarks to the Author:

The manuscript focuses on solving the groundwater depletion issues while achieving food security. I have started reading the manuscript with much interest, however, I have not found it well enough to be published in Nature Communications in its present form and content. The authors computed different scenarios based on the model they formulated, validation of these approaches are major drawback at present. My concerns are mentioned below:

- There are multiple assumptions and constraints associated with the methodology presented without proper validation of these approaches. Please validate the output based on site-scale or remote sensing approaches, otherwise it is very difficult to judge the output presented.
 - To compute the fraction of daily rainfall to be available to crop, the authors used a constant with values of 70%. I have doubts on using the static constant value during the entire Kharif season. For example, during the start of the monsoon most of the precipitated water would infiltrate and when the soil is saturated enough most fraction of rainfall would be lost as runoff. Use of static constant undermine the reality here.
 - The complete derivation of equation 8 is necessary here. Please show them in supplementary information at least.
 - The authors stated "We assume that 15% is the percentage of average annual precipitation that recharges groundwater, a reasonable assumption for subhumid to humid regions". [Lines 746-748] This is not always true. For instance, a substantial portion of India receives >2000 mm of annual rainfall and assuming 15% annual recharge, annual recharge can reach >300 mm (Asoka et al., 2017). The rate is overestimating the recharge rates observed in parts of central and southern India as recharge is strongly dependent upon subsurface properties. Asoka, A., Gleeson, T., Wada, Y. and Mishra, V., 2017. Relative contribution of monsoon precipitation and pumping to changes in groundwater storage in India. *Nature Geoscience*, 10(2), pp.109-117.
 - "The "Green Water" scenario considers no irrigation. Here, the model has only area, location, food security and nutritional constraints. We assumed no irrigation potential for the country, i.e., $\eta\% = 0$ for all the crops and districts. For the "Blue Water" scenario, 1190 constraints (586 district area constraints; 586 district irrigation constraints; 12 production constraints; six nutritional constraints – energy, proteins, fat, iron, niacin and folate) result. For the "Green Water" scenario, 604 constraints (586 district area constraints; 12 production constraints; six nutritional constraints – energy, proteins, fat, iron, niacin and folate) need to be satisfied." [Lines 776-782]
- I have little confidence on using nutritional constraints in district-level analyses, when the districts are not self sufficient in terms of food production. Import and export play crucial roles here. Different types of crops are grown in different areas based on the water availability (either irrigation or rainfall), soil type (presence of organic matter, nutrients etc.), for example, cotton is grown in black soil dominated central India and rice mostly grown in the areas of relatively rainfall i.e. high water availability (Figure 1). Estimating nutrient information for a particular district's population and extrapolating that to crop production in the same district has some flaws I believe. There will be definitely imports and export of crops between districts based on the above mentioned facts.
- Figure 3: Recommendation of 75-100% increase of rice production in eastern Madhya Pradesh in green water domain is questionable as rainfall amount is not too high to support the crop water requirement.
 - I observe the absence of citation of recent manuscripts focusing on the topic in India. Please cite some of the current studies on this aspect in India.
 - Figure S1: Depth to water table: Is this map showing data for summer time? I believe, average depth of the groundwater table would be much shallower in monsoon time.

Reviewer #2:

Remarks to the Author:

This is an important article that makes a valuable contribution but needs quite a bit of "heavy editing" before it can be acceptable.

It argues that essentially the PDS targets for crop production, even under the "zero irrigation" scenario, with a positive impact on national net farm revenue. This IS truly quite a claim.

This occurs primarily by moving rice from the dry groundwater irrigated Northwest and Indo-Gangetic states of Punjab/Haryana to the wetter North Eastern/ Southern states, by drastically reducing PDS rice procurements in those regions and switching to less water-intensive cereals in those states. The paper further argues that these "optimal" cropping patterns are broadly consistent with the cropping patterns that existed in the Northwest and Indo-Gangetic Plains before the Green Revolution and before the current Government of India PDS procurement strategy.

1. Structure of the article

Starting on Line 103 the authors state that "Three main solution directions, that cover a range of institutional and spatial scales, were considered to alleviate the current rapid groundwater decline"

I think the inclusion of previous work in this paper is distracting. I would include all previous efforts (drip, solar irrigation, feeder separation or whatever) in the literature review and not get into details of a couple that the authors have looked at in the past. It's too distracting.

I recommend getting over all these between line 72 and 74 and refocusing the paper to make the case for why a focus on the PDS system is novel and important.

These do not then need to be repeated in the Research Context – which is really just about the PDS system.

2. I found the terms "blue water" and "green water" not indicative of the scenarios. After all the blue water scenario also involves some green water use by the crops of interest, because irrigated crops also do use rainwater.

Why not call them "Irrigation Capped" and "Irrigation Zero" instead? The scenario names would then be exactly indicative of the assumptions they entailed.

There are a few things I struggled with understanding.

First – I would love to see a spatial map of where ag revenues increased and where they decreased spatially. Figure 5 in other words needs to be a map. The text mentions 4 states -- Maharashtra (111 billion INR), Punjab (79 billion INR), Karnataka (75 billion INR), 334 and Haryana (21 billion INR).

This seems fair. These are dry states that grow water intensive stuff. Why suddenly switch to a bar chart to show this?

Second, I think some like of map on the aridity index (or even just rainfall) would be valuable –to make the point that water intensive crops are being grown in dry places. The seasonal water deficit before irrigation and under current irrigation patterns might be useful.

Third, I found the "shadow price" discussion a little distracting and am worried that a casual reader will misinterpret it.

Note the following line starting at Line 280 "In terms of the PDS design and the local subsidy schemes, the shadow prices can help decide the additional investment in irrigation or in land redevelopment in each district in the country once the optimal solution is adopted."

The shadow price of irrigation water would be highest in a dry place. This is exactly what drives

irrigation projects in dry places, followed by price guarantees. But I thought the whole point of this paper is to focus on sustainability – essentially growing water intensive crops where the shadow price is lowest rather than bringing irrigation to where its highest.

I think the shadow price discussion is only confusing the higher-level message in the paper. I feel Figure 4 might be dropped altogether or moved to a supplement with only one paragraph referencing it.

Fourth, I couldn't easily locate the assumptions on yields and prices for different crops. Does the MSP have to be common across the country? Would a climate sensitive MSP work – with arid regions having higher MSPs for coarse cereals.

Suggesting this scenario because of the current highly sensitive political context of the recent farm riots, where suggesting a removal of all MSP to be replaced by private contracts is at the centre for the farm protests.

I am wondering if a scenario presented which keeps the current MSP system but improves income for farmers in the protesting states, while addressing GW sustainability would be informative.

The authors mention that farmers are open to reverting to coarse cereals but not vegetables with fluctuating prices. Given this perhaps presenting a scenario where farm incomes do not decline in the crucial states is key.

Reviewer #3:

Remarks to the Author:

Solving Groundwater Depletion in India while achieving Food Security

N. Devineni et al.

The authors, using an optimization model, obtain optimal cropping patterns for various crops in India. They identify spatial shifts in cropping patterns that will reduce groundwater depletion and enhance food security. Especially the scenario (intuitive but nice to see it come out from the optimization model) of moving more of the water intensive Rice production to north and east where water is in surplus. This result will have significant impact on agriculture policies, groundwater sustainability and food security, more so as climate changes.

I support the study. However, before I recommend acceptance I have the following comments that the authors should address.

1. The optimization has net revenue as the objective function. While this is a good proxy for yield, will the results be similar if yield is used instead? Or groundwater sustainability?
2. Figure 3 is interesting. The rice production is showing a decrease over Uttar Pradesh and Orissa, which get more rainfall and are in the Gangetic and Mahanadi River plains. Any thoughts?
3. Also surprised is that the optimization model does not shift rice production to the western Ghats region which also receive significant rainfall as the north east.

4. Rice production is suggested to shift to North East. However, this region is hilly and mountaneous and hard to reach in terms of transportation. From a carbon foot print, yes, there is stemming of groundwater depletion in Punjab/Haryana and elsewhere but the carbon foot print might be larger in enabling tillable land for rice in north east and transportation. Can the optimization model incorporate these constraints/tradeoffs?
5. Is ithe optimization performed over the entire historical period of rainfall or based on climatological rainfall? It is not clear.
6. The objective function is a single function over the entire country. That said, will the farmers/States regionally experience reduced revenue when cropping patterns are shifted? E.g., if rice is moved from Punjab/Haryana, Andhra Pradesh, Tamil Nadu and replaced with oilseeds, cereals, will get same revenue?
Also the irrigation infrastructure needs to be repurposed?
7. Following on the above, can you show the district level revenue under the two scenarios? I think it would be very interesting to see which states/region will see revenue declines from present patterns, so that policy makers can devise appropriate mitigation strategies.
8. How sensitive are these patterns to power subsidies? nationally and regionally? This can help policy makers with
9. Have the authors considered generating these optimal scenarios under reduced rainfall, such as the case that is projected during a warmer climate. This will indicate the robustness of these patters to climate fluctuations.
10. It seems like a multi-objective optimization approach with tradeoffs will be an excellent extensions to provide granular (no pun intended) options to policy makers. Thoughts/discussion on this would be helpful.
11. Lastly, India is considering revising the minimum support price (the cause of recent farmers' protest). Any thoughts on how this new law might impact the cropping patterns?

REVIEWER COMMENTS

Reviewer #1 (Remarks to the Author)

The manuscript focuses on solving the groundwater depletion issues while achieving food security. I have started reading the manuscript with much interest, however, I have not found it well enough to be published in Nature Communications in its present form and content. The authors computed different scenarios based on the model they formulated, validation of these approaches are major drawback at present. My concerns are mentioned below:

- There are multiple assumptions and constraints associated with the methodology presented without proper validation of these approaches. Please validate the output based on site-scale or remote sensing approaches, otherwise it is very difficult to judge the output presented.

Thanks for the comment. We verified our deficit model with actual groundwater depth data in several districts in Punjab where reliable historical data on the aquifers being exploited was available. Examples from three districts are shown below. The figure presents the time series comparison of groundwater depth (blue line) and cumulative deficit as computed from our model based on crops grown in that year in that district. As we emphasize, the cumulative deficit is a good proxy for groundwater extraction, and also for yield shortfall if it is not made up by irrigation. From the figures below we can see a close correspondence of the computed cumulative deficit to actual groundwater depletion at the district level. Note that these are completely independent data and we have not calibrated our model to generate these comparisons.

- To compute the fraction of daily rainfall to be available to crop, the authors used a constant with values of 70%. I have doubts on using the static constant value during the entire Kharif season. For example, during the start of the monsoon most of the precipitated water would infiltrate and when the soil is saturated enough most fraction of rainfall would be lost as runoff. Use of static constant undermine the reality here.

The reviewer is correct in suggesting that the fraction of daily rainfall available to the crop is a bulk parameter. Since the modeling unit is the district, and we compute the aggregate deficit over the season on a daily time step, a spatial averaging principle is necessary. Initial validation experiments from our previous studies were used to derive this number. The historical crop yields were related to the estimated cumulative deficits using this fraction, and this also provides an effective calibration or validation of the choice of the fraction. We have included a sensitivity analysis of this choice to the predicted changes in yields (which is part of the objective function

coefficient) in the supplemental material. Figure S2 shows the robustness of the solutions to these changes. On average, across most crops, the model solution does not change for up to a 10-25% variation in this parameter - which covers a pretty wide range. It is worth keeping in mind that the daily crop water requirement varies over the season, increasing from early in the season to a peak and then declining at the end of the season. Thus, the sensitivity to this parameter can be limited since early in the season, the potential deficit is likely to be small unless there is no rain, and later in the season, if the rains are good the deficits are small, but if the rains are weak and there is sensitivity to the threshold choice. Since the aggregate seasonal cumulative deficit is of interest, it is really dominated by the statistics of dry spells or weak rain spells and not by the large rainfall events.

- The complete derivation of equation 8 is necessary here. Please show them in supplementary information at least.

Equation 8 is the objective function coefficient.

Objective Function: The goal is to maximize the expected net national agricultural revenue

$$O = E_t \left[\sum_{j=1}^{n_d} \left(\begin{aligned} & \left(\sum_{i=1}^{n_c} \delta_{i,j} * (MSP_{i,j} - CP_{i,j}) * Y_{i,j,t} * a_{i,j} \right) \\ & - CI_j * \left(\sum_{i=1}^{n_c} \left(\left(\frac{1}{\beta_i} \right) * \eta_{i,j} * SD_{i,j,t} * \delta_{i,j} * a_{i,j} \right) * \psi_1 * g * h_j * \left(\frac{1}{\mu_p} \right) * \psi_2 \right) \right) \right] \end{aligned} \right)$$

It has two terms (the profit terms as computed by MSP – CP and the cost of irrigation terms).

The first term $\left(\sum_{i=1}^{n_c} \delta_{i,j} * (MSP_{i,j} - CP_{i,j}) * Y_{i,j,t} * a_{i,j} \right)$ is the profit (in units of INR/quantity) multiplied by crops production (yield * area). The delta function is 1 if the crop can be grown in that district and 0 if otherwise. As indicated in the paper, this decision is based on whether or not the crop has been grown in that district as evident from previous agricultural census records.

The second term computes the cost of irrigating through groundwater as the product of the unit cost of irrigation per KWH multiplied by the total estimated energy use (calculated as the water used for irrigation multiplied by the lift of that water and divided by the pump efficiency). This serves to “price” the groundwater mining /use under different scenarios.

The two terms summed together provide the net revenue to the farmers summed over all the years in the climate scenario. Thus, the expected value of the net farm revenue is maximized, consistent with the goal that the crops be procured such that food and nutrition requirements at the national level for the public distribution system are met while meeting groundwater sustainability constraints at the district level, and delivering the highest benefit to the farmers. We note that other expenses (e.g., livestock use, maintenance etc.) that are not used as part of cost estimation would not come in here.

The details of each term in the equation are explained in the methods section between lines 747 and 774.

- The authors stated "We assume that 15% is the percentage of average annual precipitation that recharges groundwater, a reasonable assumption for subhumid to humid regions". [Lines 746-748] This is not always true. For instance, a substantial portion of India receives >2000 mm of annual rainfall and assuming 15% annual recharge, annual recharge can reach >300 mm (Asoka et al., 2017). The rate is overestimating the recharge rates observed in parts of central and southern India as recharge is strongly dependent upon subsurface properties.

Asoka, A., Gleeson, T., Wada, Y. and Mishra, V., 2017. Relative contribution of monsoon precipitation and pumping to changes in groundwater storage in India. *Nature Geoscience*, 10(2), pp.109-117.

We agree with the reviewer's comment that that this is a crude estimate of the recharge rate. We chose this as a default value based on the authors' personal field experience with field recharge rates observed in a number of climate/soil settings. In our experience at field sites in alluvial and hard rock areas, the average annual recharge rates are typically 5-20% of the mean annual rainfall. This is consistent with Sukhija et al (1996)

<https://link.springer.com/article/10.1007/s100400050089> who provide estimates for much of India using tracer data, which is arguably one of the more reliable field methods. The lowest rates are in the extremely arid regions such as Rajasthan, where the rainfall is so small that the volumetric differences by using a smaller % recharge rate are not significant, and the highest values ~ 20% are in a narrow belt in Western India. In most of the other areas they report values in the 10-15% range. Another paper (Rangaran & Athavale, 2000) also provides estimates of recharge rates across India using Tritium dating

https://www.sciencedirect.com/science/article/pii/S0022169400002390?casa_token=QCFa4ji1JewAAAAA:Q1ojGTio7gWz-r2zOa6oVkY-G2xXA8_CBAg7is5jD5udoVAmXk8178uQXrHx1RK_3yeND162YBo and report very similar results (10-20% of rainfall, with a few arid places around 5%). Considering these field results the average recharge rate in Asoka et al (2017) appears to be much higher than reported in the field based estimates.

To specifically address the reviewer's comment, we also compared the mean annual recharge data provided in Asoka et al. (2017) based on PCR-GLOBWB model with our 15% of annual average rainfall estimate (see figure below). Our calculations are at a district level while those in Asoka et al are at a higher resolution - yet appear to be rather homogeneous at the district scale. We tend to have a similar spatial pattern of recharge, but the total estimated recharge is uniformly lower than in Asoka et al. This means that effectively our policy parameter for average recharge poses a more stringent constraint on what is considered a renewable supply than the estimate in Asoka et al, rather than overestimating the recharge relative to Asoka et al -- which is the concern voiced by the reviewer.

In our context, increasing the effective recharge rate in the humid regions in the country to 20% or higher to match Asoka et al, would increase the renewable water endowment for those regions. This will not impact the solution of our model since the renewable recharge constraint is typically not binding in those regions in any case, and the same crops will be shifted to those locations as they are now. We believe that given the field data, our choice is a reasonable policy level constraint for the model.

Moreover, the depth to groundwater aquifers tapped in different parts of India vary significantly, and in most of the alluvial settings the deeper aquifers are separated from the shallow upper aquifers by medium to thick impermeable layers, while in other hard rock areas, recharge occurs to fractured networks in the aquifer system. A detailed characterization of these recharge rates to the aquifers being actually used while not trivial, needs readily available data. Uncertainties in these numbers are also typically high, and age dating of the aquifers reveals ages from 1 to 10,000 years in both the sedimentary and the hard rock aquifers depending on the strata sampled, reflecting a high degree of heterogeneity and hence uncertainty.

It is important to note that our assumption applies to a long-term recharge rate. In our model we use this fraction (15%) simply to put a bound on what is considered to be renewable groundwater. Effectively this is a policy parameter that prescribes a sustainable use regime by putting a limit on how much can be withdrawn on a renewable basis. It does not directly enter into a mass balance calculation for groundwater dynamics. One can vary this threshold regionally if needed, but that adds additional assumptions that may or may not be easy to develop or defend.

- "The "Green Water" scenario considers no irrigation. Here, the model has only area, location, food security and nutritional constraints. We assumed no irrigation potential for the country, i.e., $\eta_{i,j} = 0$ for all the crops and districts. For the "Blue Water" scenario, 1190 constraints (586

district area constraints; 586 district irrigation constraints; 12 production constraints; six nutritional constraints – energy, proteins, fat, iron, niacin and folate) result. For the “Green Water” scenario, 604 constraints (586 district area constraints; 12 production constraints; six nutritional constraints – energy, proteins, fat, iron, niacin and folate) need to be satisfied.” [Lines 776-782] I have little confidence on using nutritional constraints in district-level analyses, when the districts are not self sufficient in terms of food production. Import and export play crucial roles here. Different types of crops are grown in different areas based on the water availability (either irrigation or rainfall), soil type (presence of organic matter, nutrients etc.), for example, cotton is grown in black soil dominated central India and rice mostly grown in the areas of relatively rainfall i.e. high water availability (Figure 1). Estimating nutrient information for a particular district's population and extrapolating that to crop production in the same district has some flaws I believe. There will be definitely imports and export of crops between districts based on the above mentioned facts.

The reviewer is correct in pointing out that different crops are grown in different places so it is impossible to derive the nutritional needs of the population from crop yields for each district. As it follows, the nutrition constraint used in the model is not at the district-level. We estimate the nutrition level required for the entire country to see if the national crop production can satisfy this or not. The nutrition constraint thus targets the national public distribution system and not any district level needs. This point has been clarified in the text further.

- Figure 3: Recommendation of 75-100% increase of rice production in eastern Madhya Pradesh in green water domain is questionable as rainfall amount is not too high to support the crop water requirement.

Thank you for the observation. The authors would like to point out that Madhya Pradesh has shallow ground water and reasonable rainfall with around 40% irrigated area. The percentage increase in rice has to be viewed in the context of the current total production level. In absolute terms this is not a dramatic increase in land area and is a consequence of the relative values of different crops, the water availability and productivity across all regions.

- I observe the absence of citation of recent manuscripts focusing on the topic in India. Please cite some of the current studies on this aspect in India.

We have added some recent literature that more closely relates to our work. We are happy to add any others that the reviewers may find relevant, keeping in mind the total number of citations limit for the journal.

The follow citations are added in the revised manuscript.

1. Mishra, V., Asoka, A., Vatta, K., & Lall, U. (2018). Groundwater depletion and associated CO2 emissions in India. *Earth's Future*, 6(12), 1672-1681.
2. Fishman, R. Groundwater depletion limits the scope for adaptation to increased rainfall variability in India. (2018). *Climatic Change* 147, 195–209. <https://doi.org/10.1007/s10584-018-2146-x>
3. Bhogal, S., & Vatta, K. (2021). Can crop diversification be widely adopted to solve the water crisis in Punjab?. *CURRENT SCIENCE*, 120(8), 1303.
4. Asoka, A., Gleeson, T., Wada, Y. and Mishra, V. (2017), Relative contribution of monsoon precipitation and pumping to changes in groundwater storage in India. *Nature Geoscience*, 10(2), pp.109-117.

- Figure S1: Depth to water table: Is this map showing data for summer time? I believe, average depth of the groundwater table would be much shallower in monsoon time.

This data is from Central Groundwater Board and is an average depth over pre- and post-monsoon.

Reviewer #2 (Remarks to the Author)

This is an important article that makes a valuable contribution but needs quite a bit of "heavy editing" before it can be acceptable.

It argues that essentially the PDS targets for crop production, even under the "zero irrigation" scenario, with a positive impact on national net farm revenue. This IS truly quite a claim.

This occurs primarily by moving rice from the dry groundwater irrigated Northwest and Indo-Gangetic states of Punjab/Haryana to the wetter North Eastern/ Southern states, by drastically reducing PDS rice procurements in those regions and switching to less water-intensive cereals in those states.

The paper further argues that these "optimal" cropping patterns are broadly consistent with the cropping patterns that existed in the Northwest and Indo-Gangetic Plains before the Green Revolution and before the current Government of India PDS procurement strategy.

1. Structure of the article

Starting on Line 103 the authors state that "Three main solution directions, that cover a range of institutional and spatial scales, were considered to alleviate the current rapid groundwater decline" I think the inclusion of previous work in this paper is distracting. I would include all previous efforts (drip, solar irrigation, feeder separation or whatever) in the literature review and not get into details of a couple that the authors have looked at in the past. It's too distracting. I recommend getting over all these between line 72 and 74 and refocusing the paper to make the case for why a focus on the PDS system is novel and important. These do not then need to be repeated in the Research Context – which is really just about the PDS system.

We have updated this part based on the suggestion. These paragraphs are significantly shortened (Lines 110-124 and 133-135 are removed) and to the point now while citing our previous published studies where past work has been mentioned.

2. I found the terms "blue water" and "green water" not indicative of the scenarios. After all the blue water scenario also involves some green water use by the crops of interest, because irrigated crops also do use rainwater. Why not call them "Irrigation Capped" and "Irrigation Zero" instead? The scenario names would then be exactly indicative of the assumptions they entailed.

Thank you for pointing it out. We changed the terminology throughout the text and in the figures it in line with your suggestion.

There are a few things I struggled with understanding.

First – I would love to see a spatial map of where ag revenues increased and where they decreased spatially. Figure 5 in other words needs to be a map. The text mentions 4 states -- Maharashtra (111 billion INR), Punjab (79 billion INR), Karnataka (75 billion INR), 334 and Haryana (21 billion INR). This seems fair. These are dry states that grow water intensive stuff. Why suddenly switch to a bar chart to show this?

Thank you for the suggestion. We have converted Figure 5 to maps. It is now Figure 4. For revenue, we show 3 categories - negative, neutral, and positive. Based on this, we find that much of country falls in the neutral category, while some states (districts) are negative and some are positive, as we had highlighted in the original manuscript. Based on the revised assessment and maps, we have also updated the discussion in the “Redistribution of Income, Water and Energy Use from Changing PDS” section including other aspects (water saved, energy saved and opportunity cost).

Second, I think some like of map on the aridity index (or even just rainfall) would be valuable – to make the point that water intensive crops are being grown in dry places. The seasonal water deficit before irrigation and under current irrigation patterns might be useful.

Thank you. Actually, we have the rainfall and coefficient of variation of rainfall map presented in the supplement (Figure S1). Based on your suggestion, the maps for water savings calculated is also shown in Figure4 second panel.

Third, I found the “shadow price” discussion a little distracting and am worried that a casual reader will misinterpret it. Note the following line starting at Line 280 “In terms of the PDS design and the local subsidy schemes, the shadow prices can help decide the additional investment in irrigation or in land redevelopment in each district in the country once the optimal solution is adopted.” The shadow price of irrigation water would be highest in a dry place. This is exactly what drives irrigation projects in dry places, followed by price guarantees. But I thought the whole point of this paper is to focus on sustainability – essentially growing water intensive crops where the shadow price is lowest rather than bringing irrigation to where its highest. I think the shadow price discussion is only confusing the higher-level message in the paper. I feel Figure 4 might be dropped altogether or moved to a supplement with only one paragraph referencing it.

Thank you very much for your insightful comment. We have reviewed that section, and we agree that the water constraint shadow price discussion could be distracting for the readers given the focus of the paper. We have therefore moved the entire discussion to the supplement.

Fourth, I couldn't easily locate the assumptions on yields and prices for different crops. Does the MSP have to be common across the country? Would a climate sensitive MSP would work – with arid regions having higher MSPs for coarse cereals. Suggesting this scenario because of the current highly sensitive political context of the recent farm riots, where suggesting a removal of all MSP to be replaced by private contracts is at the centre for the farm protests. I am wondering if a scenario presented which keeps the current MSP system but improves income for farmers in the protesting states, while addressing GW sustainability would be informative.

MSP is currently fixed across the country, and hence was the basis for our model. It is indeed a

smaller perturbation of the current system compared to a spatially variable MSP as proposed by the reviewer.

An MSP that was indexed to aridity would arise naturally if it was tied to cost of cultivation (which it is to an extent) and electricity and other subsidies for production (that are then not accounted for in the cost of cultivation) were not offered. We considered such a run, but finally chose to stay with the fixed MSP, since a) it was unclear from the field level studies and discussions that the subsidies offered by the states could be changed at the national level, b) there were concerns of perception of inequity if different regions were given different MSPs, and especially in the same state (where larger states may cover several climate zones), and then parsing the results would be a bit more challenging.

As to the second point of reviewer for improving the income for farmers in the states that would be losing under the proposed change - this is an important point and we have considered the following: 1) First, we documented the net loss of revenue the farmers would experience relative to the current situation; 2) Second, we considered the value of the savings to the state from electricity subsidy due to the crop switch; 3) Third, we considered the change in the MSP for the substitute crops that would allow a package to be created (together with the savings from the electricity subsidy) that would make the situation revenue neutral for the farmers. We find that coordinating such a system could still be challenging.

The authors mention that farmers are open to reverting to coarse cereals but not vegetables with fluctuating prices. Given this perhaps presenting a scenario where farm incomes do not decline in the crucial states is key.

Yes, we agree. De-risking the price fluctuations of vegetables and fruits and higher cash value crops is a key criterion that needs to be addressed in these states. Past efforts at contract farming have provided the proof of concept but has faced implementation issues due to renegeing of contracts. Drawing from field studies, we have highlighted some publications (also referred in the manuscript) by one of the authors that addresses these issues in further detail.

- Huh, W. T., & Lall, U. (2013). Optimal crop choice, irrigation allocation, and the impact of contract farming. *Production and Operations Management*, 22(5), 1126-1143.
- Huh, W. T., Athanassoglou, S., & Lall, U. (2012). Contract farming with possible renegeing in a developing country: Can it work?. *IIMB Management Review*, 24(4), 187-202.
- Federgruen, Awi, Upmanu Lall, and A. Serdar Şimşek. (2019). Supply chain analysis of contract farming. *Manufacturing & Service Operations Management* 21(2), 361-378.

Though we have added some discussion in the text (“Redistribution of Income, Water and Energy Use from Changing PDS” Section), we think that adding the content on revenue guaranteeing ideas to the current paper as would likely shift the focus of this paper and make it more complex.

Reviewer #3 (Remarks to the Author)

Solving Groundwater Depletion in India while achieving Food Security

N. Devineni et al.

The authors, using an optimization model, obtain optimal cropping patterns for various crops in India. They identify spatial shifts in cropping patterns that will reduce groundwater depletion and enhance food security. Especially the scenario (intuitive but nice to see it come out from the optimization model) of moving more of the water intensive Rice production to north and east where water is in surplus. This result will have significant impact on agriculture policies, groundwater sustainability and food security, more so as climate changes.

I support the study. However, before I recommend acceptance I have the following comments that the authors should address.

1. The optimization has net revenue as the objective function. While this is a good proxy for yield, will the results be similar if yield is used instead? Or groundwater sustainability?

Thank you for the comment. The total net farm revenue with the constraints on nutrition and meeting the target production of each crop ensure that the procurement goals are met while delivering the highest net revenue to the agricultural sector. This will happen by a selection of the locations that give the highest yield to each group while meeting a target groundwater constraint. So, the sustainability, economic development and yield objectives are simultaneously met by this strategy. As the reviewer has suggested, one could vary the procurement target and allowable groundwater use targets parametrically to evaluate how their level changes the cropping pattern. Our sensitivity analyses indicate that the general crop shift pattern is quite stable to reasonable perturbations in these constraints.

2. Figure 3 is interesting. The rice production is showing a decrease over Uttar Pradesh and Orissa, which get more rainfall and are in the Gangetic and Mahanadi River plains. Any thoughts?

Thank you for the observation. We would like to point out three underlying reasons that would help explain the fact. First, it is important to keep in mind that the Figure shows % changes from the current cropping pattern. So, areas where there is a high land area devoted to a particular crop may still experience a modest % decline, while areas for which the area under crop production is already relatively small, may show a high % increase (as seen in the figure), even if in absolute terms the amount of increase is small. Second, the productivity of each district is also considered in our analysis. Third, we account for the daily precipitation variability, i.e., for two places with the same total annual rainfall, the one with higher variability (or dry spells) the corresponding decline in yield or irrigation requirement is recognized. This is not the case if average seasonal or average daily rainfall is used in the analysis. Orissa experiences high rainfall but with very high daily amounts (often associated with tropical cyclones) that cannot be fully utilized by the crop, as it is interspersed with persistent dry spells. For instance, parts of UP that show a modest increase in rice are the regions near the Himalayas where the rainfall patterns are more consistent, whereas the regions that show a decline are the ones with high intermittence and

typically lower rainfall, and with lower overall recorded productivity as well.

3. Also surprised is that the optimization model does not shift rice production to the western Ghats region which also receive significant rainfall as the north east.

Recall that the figure shows % **changes** from the current situation, and that the total crop area and irrigated crop area per district are constrained to current levels. Currently, rice is already grown in the western ghats (% cropped area for rice is more than 75%). Under rainfed conditions thus, the model shows only shows a modest increase from current level of land already allocated to rice.

4. Rice production is suggested to shift to North East. However, this region is hilly and mountaneous and hard to reach in terms of transportation. From a carbon foot print, yes, there is stemming of groundwater depletion in Punjab/Haryana and elsewhere but the carbon foot print might be larger in enabling tillable land for rice in north east and transportation. Can the optimization model incorporate these constraints/tradeoffs?

Thank you for the observation. Traditionally rice consumption is much higher in the Northeast, East and in the South than in the Punjab/Haryana area, so the current procurement system already entails a significant transportation requirement out of the North to the areas that consume more rice. We were aware of this issue as we developed the model. As a proxy, we have limited the cropping area and irrigated area to currently suitable areas and no new land is being used or shown for crop production by the model. The model considers only substitution of crops in currently tilled land, and this constraint is applied at the district level.

5. Is the optimization performed over the entire historical period of rainfall or based on climatological rainfall? It is not clear.

It is over the entire historical period (1901-2010) based on daily water deficit and expected yield in light of the cumulative water deficit per crop season.

6. The objective function is a single function over the entire country. That said, will the farmers/States regionally experience reduced revenue when cropping patterns are shifted? E.g., if rice is moved from Punjab/Haryana, Andhra Pradesh, Tamil Nadu and replaced with oilseeds, cereals, will get same revenue? Also the irrigation infrastructure needs to be repurposed?

We show this change in revenue in Figure 5. Existing irrigation infrastructure will be used for the new crops. We do not consider a change in the irrigation infrastructure – while being predominantly flood based across the country, much of the discussion around the irrigation infrastructure would be rather complex given the focus of the paper. In reality, there is a potential for increased irrigation efficiency nonetheless as the crops change. Note that we focus strictly on redesigning the PDS and restrict to those crops, rather than considering all possible crops. Cash crops can be much more remunerative, especially if they are de-risked from price fluctuations as is the case with PDS crops. The issue with PDS is that the procurement is targeted to certain locations for certain crops and is not applied in a uniform way for each crop across the country.

7. Following on the above, can you show the district level revenue under the two scenarios? I think it would be very interesting to see which states/region will see revenue declines from present patterns, so that policy makers can devise appropriate mitigation strategies.

Thank you for your suggestion. We now have a map (see Figure 4) that shows it. The discussion for the figure is also updated in the text (lines 256-271).

8. How sensitive are these patterns to power subsidies? nationally and regionally? This can help policy makers with

The sensitivity analysis of reduction of net revenue per crop by 10 to 50% parametrically was conducted for some insight into this question. The idea was that if the subsidy were removed the net revenue for that crop would decrease. We find that the general pattern of crop allocation is robust to these changes especially for rice, the crop of concern. In a sense the rainfed solution and the irrigated solution with irrigation limited to the current irrigated area, provide bounds to the solution where new irrigation infrastructure is not considered.

9. Have the authors considered generating these optimal scenarios under reduced rainfall, such as the case that is projected during a warmer climate. This will indicate the robustness of these patterns to climate fluctuations.

We have not done it in this study, but we acknowledge that it can be done. In fact, we are working on a paper that plans for food security under climate change, and also in using this for annual planning. There continue to be significant uncertainties with climate change projections, but the consensus forecasts for India tend to be slightly wetter, and lack spatial specificity or robustness.

10. It seems like a multi-objective optimization approach with tradeoffs will be an excellent extension to provide granular (no pun intended) options to policy makers. Thoughts/discussion on this would be helpful.

Multi-objective optimization is a computationally effective way to explore and discuss results that can be obtained with any one of the objectives, while varying the others parametrically as constraints. We find that both can be a useful formulation, but often it is easier to communicate the results with respect to a meaningful objective to decision makers and expose the implications for the other objectives.

11. Lastly, India is considering revising the minimum support price (the cause of recent farmers' protest). Any thoughts on how this new law might impact the cropping patterns?

This is a very broad issue at this point, since many ideas are being discussed, including opening up the market to the private sector. Past and current attempts at stimulating contract farming were in the same spirit. We anticipate exploring a few of the directions that can emerge here since this is an important design issue. The reduced costs from our model for each variable can to an extent inform the question of what the implications would be, but it would be better to explore each specific proposal in some depth, which is beyond the scope of the current paper.

Reviewers' Comments:

Reviewer #1:

Remarks to the Author:

The authors have made some changes on the manuscript during this revision, I still have the following concerns on the paper:

1. I am not satisfied with the validation approach provided in the response. The authors attempted to show a simple validation approach using the groundwater table depth with cumulative deficit. The plot of cumulative deficit follows a smooth sigmoid function without any change, while the groundwater level has lots of deviation in these three districts. The relationship is okay to show in the districts where groundwater level is continuously falling, this would not work in districts with increasing or neutral long-term groundwater level conditions.

2. I am not satisfied either with the responses provided by the authors on complete derivation or explanation of the equation 8. As the authors have not cited any literature here, it is considered that the authors have derived this equation on their own, where each term must be critically checked unless the result can be erroneous. The objective function section (Lines 724-752) has a lot of assumptions that can vary much across the spatial as well as temporal scale across India. The logic behind the consideration of all these terms in equation 8 needed to be justified.

3. Regarding the groundwater recharge rate: the authors have not properly answered this concern behind their consideration of groundwater recharge as the 15% of annual rainfall. I referred Asoka et al. (2017) to indicate groundwater recharge from the CGWB well data (Figure 4a in that paper). The authors have not responded to that rather they provided the modeled recharge from that article which follows precipitation only without considering geology, irrigation return flow etc. The original rate of recharge is much lower in west coast and parts of the central India (eg. Madhya Pradesh) (Figure 4a of Asoka et al., 2017) than the precipitation guided modeled approach.

Finally, the manuscript considers very simplistic approach to derive the country-wide stats without considering the spatial as well as the temporal variabilities of each of these model parameters used. This leads to the results look uncertain and questionable once you change the value of one parameter.

Reviewer #3:

Remarks to the Author:

I have closely read the response to comments and the revised manuscript. The authors have put in lot of efforts in robustly responding to all the comments (including mine). The revised manuscript makes an important contribution to India's agriculture policy making. I am happy to recommend acceptance.

REVIEWER COMMENTS

Reviewer #1 (Remarks to the Author):

The authors have made some changes on the manuscript during this revision, I still have the following concerns on the paper:

1. I am not satisfied with the validation approach provided in the response. The authors attempted to show a simple validation approach using the groundwater table depth with cumulative deficit. The plot of cumulative deficit follows a smooth sigmoid function without any change, while the groundwater level has lots of deviation in these three districts. The relationship is okay to show in the districts where groundwater level is continuously falling, this would not work in districts with increasing or neutral long-term groundwater level conditions.

All models at the scale considered require some assumptions. This can be seen in the large-scale water models by Yoshi Wada and his group, as well as the papers by Ximing Cai that consider detailed regional or country-scale modeling, as well as in other papers that consider regional to global hydrologic scales. As an example, typically, some soil maps are used and the assumptions are that a) these actually represent the soils in the region, and b) that the textbook values of the soil parameters are applicable. These variations are then specified, and there is no ability to really validate these assumptions in a spatially distributed manner. The same is true for the primary hydrologic driver -- precipitation -- a field that is highly variable in space and time, but gridded values in space and time that are inferred from a rather sparse spatial network or from proxies such as satellites or radar are used, and assumed to be representative. Indeed, the very popular VIC model recognizes that sub-scale spatial variability in hydraulic parameters needs to be parameterized in some way. However, in the large-scale applications of even such a model there is no verification of the parameterization in a spatially distributed way - calibration exercises are undertaken and often the performance is deemed satisfactory in terms of reproduction of monthly flow at some stations, even though the model has a much higher space-time resolution. Similarly, adherents of the TOPMODEL have long made assumptions as to the exponential extinction of hydraulic conductivity in the vertical, which would effectively lead to zero recharge to the deep aquifers of interest. We are not interested in criticizing the field, rather, we want to emphasize that modeling assumptions are common and are consistent with the goals of an exercise. In this case, we used the ideas developed from normalized deficit index, a water risk index that we developed for India (<https://agupubs.onlinelibrary.wiley.com/doi/abs/10.1002/wrcr.20184>) as a means to constraint water availability and crop yield.

2. I am not satisfied either with the responses provided by the authors on complete derivation or explanation of the equation 8. As the authors have not cited any literature here, it is considered that the authors have derived this equation on their own, where each term must be critically checked unless the result can be erroneous. The objective function section (Lines 724-752) has a

lot of assumptions that can vary much across the spatial as well as temporal scale across India. The logic behind the consideration of all these terms in equation 8 needed to be justified.

The equation is a standard cost and revenue computation, and the details are presented. The comment that these parameters can vary over India is noted, and the cost of cultivation and yield parameters that are the most critical terms in the objective function were painstakingly put together from various government data sources at the district level in India.

3. Regarding the groundwater recharge rate: the authors have not properly answered this concern behind their consideration of groundwater recharge as the 15% of annual rainfall. I referred Asoka et al. (2017) to indicate groundwater recharge from the CGWB well data (Figure 4a in that paper). The authors have not responded to that rather they provided the modeled recharge from that article which follows precipitation only without considering geology, irrigation return flow etc. The original rate of recharge is much lower in west coast and parts of the central India (eg. Madhya Pradesh) (Figure 4a of Asoka et al., 2017) than the precipitation guided modeled approach.

Finally, the manuscript considers very simplistic approach to derive the country-wide stats without considering the spatial as well as the temporal variabilities of each of these model parameters used. This leads to the results look uncertain and questionable once you change the value of one parameter.

In this case, we have 2 models - one considering rainfed conditions and one considering groundwater-based irrigation. The recharge analysis does not apply to the rain fed case at all. For the second case, the estimated recharge applies as an upper bound to the amount of pumping allowed locally from a sustainability perspective. It is not used in a mass balance analysis as the reviewer may have anticipated – all it is used for is to limit the amount of irrigation allowed. This is really a policy variable since it specifies a limit that would be applied, and we have considered a fair amount of sensitivity analysis. The central point of the paper is that whether no groundwater use is allowed (rainfed) or some groundwater use is allowed (limited to some fraction of the average local rainfall), significant shifts in crops are indicated in India to achieve sustainability while increasing or maintaining net farm income across the country. The fact that this conclusion emerges with similar shift patterns irrespective of whether or not any recharge and pumping are considered obviates the criticism. Further, we provided references that showed real world experiments on recharge rates across representative areas in the country that are consistent with the range of parameters we have tested, and indicated that we made those assumptions based on our own field experience in India and interviews with agricultural specialists who have done similar experiments. Yes, one could specify variations in these parameters and re-run the model, but this will not change the main conclusions of the paper. At most this would lead to a second order effect -- again note that the rainfed model considers no recharge so that is one extreme end point.

Reviewer #3 (Remarks to the Author):

I have closely read the response to comments and the revised manuscript. The authors have put in lot of efforts in robustly responding to all the comments (including mine). The revised manuscript makes an important contribution to India's agriculture policy making. I am happy to recommend acceptance.

Thank you.

Reviewers' Comments:

Reviewer #3:

Remarks to the Author:

Solving Groundwater Depletion in India while achieving Food Security

N. Devineni et al.

I read the revised version of the manuscript and the responses to reviewer comments, especially reviewer #1.

Below, I will attempt to sort the confusion by this reviewer and provide a broader perspective.

The authors have done an excellent job in developing a well thought country scale integrated model for groundwater depletion. This while addressing the food security, land and water (rainfall and groundwater) constraints. Given the challenges in obtaining groundwater data from India, especially, and other data (food production, cropping pattern, hydrology etc.), the authors need to be commending in pulling all of this together in their proposed optimization modeling framework. Further, the optimization model is offered as a realistic representation for policy makers. It is simple enough to easily understand the issues at the national level and realistic enough that it captures all the aspects of the system - rainfall, groundwater, support price, cropping pattern, food security etc. To me, this is the *best* part of this study.

The groundwater representation in the optimization framework is conceptual and representative, in that, it assumes a fraction of rainfall for recharge and available for crop and pumping. Clearly, this is not for mass balance modeling of groundwater, as the reviewer #1 seems to think/want. The conceptual representation of groundwater is very good for capturing the space-time variability at a larger scale and for evaluating policy options.

The objective function proposed in the paper is the net national agricultural revenue. From a national policy making perspective this is a very good objective function. Of course, one can consider several interesting variations (couple of them I suggested in my earlier review) such as, multiple objectives - revenue, groundwater depletion food security etc. - and perform trade off analysis. In addition, regional revenues and groundwater depletion can also be considered to implement this framework over regions. Regardless, the objective function proposed is an 'aggregate' response variable that is quite standard in such modeling efforts. The food security and nutrition needs are placed as constraints which can be changed by policy makers to evaluate policy alternatives.

The importance and strength of this study is this research is the guidance for policy makers on shifting various crops across the country to 'optimal' locations. It is well known at an intuitive level to economists, policy makers that support price and subsidies have (and are) led to suboptimal growing of crops. The stark example is the growing of rice in north India (Punjab, Haryana etc.) where rainfall is lower and are also experiencing severe groundwater depletion - but, support prices and subsidies are incentivizing this behavior. The optimization model in this study clearly suggests moving this production to other water-surplus regions of the country while maintaining the revenue, food security etc. This study provides a good evidence of this and can enable much needed policy changes.

Since the processes such as groundwater depletion are modeled in a simpler approach, it enables robust integration with other aspects of the system. A full fledged groundwater modeling would be an entirely separately endeavor and will not be possible to incorporate in a policy scale model, like the one here. It is worthwhile to develop such a model for the country, but

it is not a limitation in this study.

Summary: The study provides a high level, yet realistic, integrated model for groundwater depletion in combination with crop production, food security, rainfall variability, revenue etc.

The results reinforce the general thinking prevalent in policy circles, but, provides a simple and robust framework to evaluate policy options. The study should be viewed in this perspective

and not as a detailed model of any individual hydrologic or economic process.

REVIEWERS' COMMENTS

Reviewer #3 (Remarks to the Author):

Solving Groundwater Depletion in India while achieving Food Security
N. Devineni et al.

I read the revised version of the manuscript and the responses to reviewer comments, especially reviewer #1. Below, I will attempt to sort the confusion by this reviewer and provide a broader perspective.

The authors have done an excellent job in developing a well thought country scale integrated model for groundwater depletion. This while addressing the food security, land and water (rainfall and groundwater) constraints. Given the challenges in obtaining groundwater data from India, especially, and other data (food production, cropping pattern, hydrology etc.), the authors need to be commending in pulling all of this together in their proposed optimization modeling framework. Further, the optimization model is offered as a realistic representation for policy makers. It is simple enough to easily understand the issues at the national level and realistic enough that it captures all the aspects of the system - rainfall, groundwater, support price, cropping pattern, food security etc. To me, this is the *best* part of this study.

The groundwater representation in the optimization framework is conceptual and representative, in that, it assumes a fraction of rainfall for recharge and available for crop and pumping. Clearly, this is not for mass balance modeling of groundwater, as the reviewer #1 seems to think/want. The conceptual representation of groundwater is very good for capturing the space-time variability at a larger scale and for evaluating policy options.

The objective function proposed in the paper is the net national agricultural revenue. From a national policy making perspective this is a very good objective function. Of course, one can consider several interesting variations (couple of them I suggested in my earlier review) such as, multiple objectives - revenue, groundwater depletion food security etc. - and perform trade off analysis. In addition, regional revenues and groundwater depletion can also be considered to implement this framework over regions. Regardless, the objective function proposed is an 'aggregate' response variable that is quite standard in such modeling efforts. The food security and nutrition needs are placed as constraints which can be changed by policy makers to evaluate policy alternatives.

The importance and strength of this study is this research is the guidance for policy makers on shifting various crops across the country to 'optimal' locations. It is well known at an intuitive level to economists, policy makers that support price and subsidies have (and are) led to suboptimal growing of crops. The stark example is the growing of rice in north India (Punjab, Haryana etc.) where rainfall is lower and are also experiencing severe groundwater depletion - but support prices and subsidies are incentivizing this behavior. The optimization model in this study clearly suggests moving this production to other water-surplus regions of the country while maintaining the revenue, food security etc. This study provides a good evidence of this and can enable much needed policy changes.

Since the processes such as groundwater depletion are modeled in a simpler approach, it enables robust integration with other aspects of the system. A full fledged groundwater modeling would be an entirely separately endeavor and will not be possible to incorporate in a policy scale model, like the one here. It is worthwhile to develop such a model for the country, but it is not a limitation in this study.

Summary: The study provides a high level, yet realistic, integrated model for groundwater depletion in combination with crop production, food security, rainfall variability, revenue etc. The results reinforce the general thinking prevalent in policy circles, but, provides a simple and robust framework to evaluate policy options. The study should be viewed in this perspective and not as a detailed model of any individual hydrologic or economic process.

Thank you very much for taking the time to provide an overall assessment and context for the article.